# On Model Calibration for Long-Tailed Object Detection and Instance Segmentation

**Tai-Yu Pan**[1][*]   **Cheng Zhang**[1][*]   **Yandong Li**[2]   **Hexiang Hu**[2]
**Dong Xuan**[1]   **Soravit Changpinyo**[2]   **Boqing Gong**[2]   **Wei-Lun Chao**[1]

[1]The Ohio State University   [2]Google Research

## Abstract

Vanilla models for object detection and instance segmentation suffer from the heavy bias toward detecting frequent objects in the long-tailed setting. Existing methods address this issue mostly during training, *e.g.*, by re-sampling or re-weighting. In this paper, we investigate a largely overlooked approach — *post-processing calibration* of confidence scores. We propose NORCAL, **Normalized Calibration** for long-tailed object detection and instance segmentation, a simple and straightforward recipe that reweighs the predicted scores of each class by its training sample size. We show that separately handling the background class and normalizing the scores over classes for each proposal are keys to achieving superior performance. On the LVIS dataset, NORCAL can effectively improve nearly all the baseline models not only on rare classes but also on common and frequent classes. Finally, we conduct extensive analysis and ablation studies to offer insights into various modeling choices and mechanisms of our approach. Our code is publicly available at https://github.com/tydpan/NorCal.

## 1   Introduction

Object detection and instance segmentation are the fundamental tasks in computer vision and have been approached from various perspectives over the past few decades [8, 13, 26, 36, 52]. With the recent advances in neural networks [1, 4, 10, 18, 29, 30, 32, 37, 40, 41, 43], we have witnessed an unprecedented breakthrough in detecting and segmenting frequently seen objects such as people, cars, and TVs [14, 15, 21, 31, 70]. Yet, when it comes to detect rare, less commonly seen objects (*e.g.*, walruses, pitchforks, seaplanes, etc.) [12, 51], there is a drastic performance drop largely due to insufficient training samples [47, 68]. How to overcome the "long-tailed" distribution of different object classes [69] has therefore attracted increasing attention lately [27, 35, 46, 55].

To date, most existing works tackle this problem in the *model training phase*, *e.g.*, by developing algorithms, objectives, or model architectures to tackle the long-tailed distribution [12, 20, 27, 46, 48, 53, 55, 57, 58]. Wang et al. [55] investigated the widely used instance segmentation model Mask R-CNN [18] and found that the performance drop comes primarily from *mis-classification of object proposals*. Concretely, the model tends to give frequent classes higher confidence scores [7], hence biasing the label assignment towards frequent classes. This observation suggests that techniques of class-imbalanced learning [2, 6, 16, 42] can be applied to long-tailed detection and segmentation.

Building upon the aforementioned observation, we take another route in the *model inference phase* by explicit *post-processing calibration* [2, 22, 23, 33, 61], which adjusts a classifier's confidence scores among classes, without changing its internal weights or architectures. Post-processing calibration is efficient and widely applicable since it requires no re-training of the classifier. Its effectiveness

---

[*]Equal contributions

35th Conference on Neural Information Processing Systems (NeurIPS 2021).

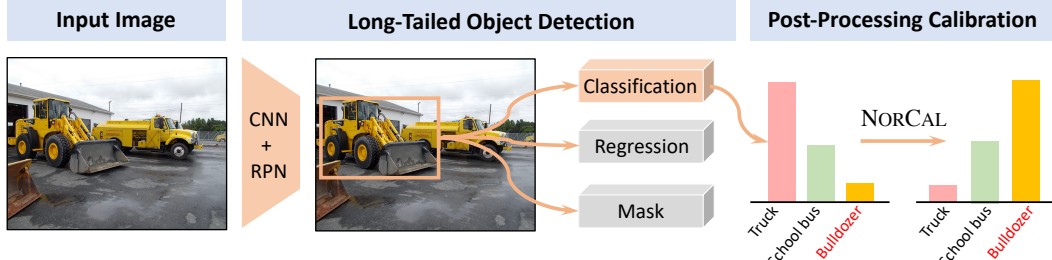

Figure 1: **Normalized Calibration (NORCAL).** Object detection or instance segmentation models (*e.g.*, [18, 43]) trained with data from a long-tailed distribution tend to output higher confidence scores for the head classes (*e.g.*, "Truck") than for the tail ones (*e.g.*, the true class label "Bulldozer"). NORCAL investigates a simple but largely overlooked approach to correct this mistake — post-processing calibration of the classification scores *after training* — and significantly improves nearly all the models we consider.

on multiple imbalanced classification benchmarks [22, 59] may also translate to long-tailed object detection and instance segmentation.

In this paper, we propose a simple post-processing calibration technique inspired by class-imbalanced learning [33, 61] and show that it can significantly improve a pre-trained object detector's performance on detecting both rare and common classes of objects. *We note that our results are in sharp contrast to a couple of previous attempts on exploring post-processing calibration in object detection [7, 27], which reported poor performance and/or sensitivity to hyper-parameter tuning.* We also note that the calibration techniques in [54, 55] are implemented in the training phase and are not post-processing.

Concretely, we apply post-processing calibration to the classification sub-network of a pre-trained object detector. Taking Faster R-CNN [43] and Mask R-CNN [18] for examples, they apply to each object proposal a $(C + 1)$-way softmax classifier, where $C$ is the number of foreground classes, and 1 is the background class. To prevent the scores from being biased toward frequent classes [7, 55], we *re-scale the logit* of every class according to its class size, *e.g.*, number of training images. Importantly, we leave the logit of the background class intact because (a) the background class has a drastically different meaning from object classes and (b) its value does not affect the ranking among different foreground classes. After adjusting the logits, we then re-compute the confidence scores (with normalization across all classes, including the background) to decide the label assignment for each object proposal[2] (see Figure 1). We note that it is crucial to normalize the scores across all classes since it triggers *re-ranking of the detection results within each class* (see Figure 3), influencing the class-wise precision and recall. Instead of separately adjusting each class by a specific factor [7], we follow [6, 33, 61] to set the factor as a function of the class size, leaving only one hyper-parameter to tune. We find that it is robust to use the training set to set this hyper-parameter, making our approach applicable to scenarios where collecting a held-out representative validation set is challenging.

Our approach, named **Normalized Calibration** for long-tailed object detection and instance segmentation (NORCAL), is model-agnostic as long as the detector has a softmax classifier or multiple binary sigmoid classifiers for the objects and the background. We validate NORCAL on the LVIS [12] dataset for both long-tailed object detection and instance segmentation. NORCAL can consistently improve not only baseline models (*e.g.*, Faster R-CNN [43] or Mask R-CNN [18]) but also many models that are dedicated to the long-tailed distribution. Hence, our best results notably advance the state of the art. Moreover, NORCAL can improve both the standard average precision (AP) and the category-independent $AP^{Fixed}$ metric [7], implying that NORCAL does not trade frequent class predictions for rare classes but rather *improve the proposal ranking within each class*. Indeed, through a detailed analysis, we show that NORCAL can in general improve both the precision and recall for each class, making it appealing to almost any existing evaluation metrics. Overall, we view NORCAL a simple plug-and-play component to improve object detectors' performance during inference.

## 2 Related Work

**Long-tailed detection and segmentation.** Existing works on long-tailed object detection can roughly be categorized into re-sampling, cost-sensitive learning, and data augmentation. *Re-sampling*

---

[2]Popular evaluation protocols allow multiple labels per proposal if their confidence scores are high enough.

*methods* change the long-tailed training distribution into a more balanced one by sampling data from rare classes more often [3, 12, 45]. *Cost-sensitive learning* aims at adjusting the loss of data instances according to their labels [19, 46, 48, 53]. Building upon these, some methods perform two- or multi-staged training [20, 22, 27, 42, 55–57, 67], which first pre-train the models in a conventional way, using data from all or just the head classes; the models are then fine-tuned on the entire long-tailed data using either re-sampling or cost-sensitive learning. Besides, another thread of works leverages data augmentation for the object instances of the tail classes to improve long-tailed object detection [9, 39, 65, 66].

In contrast to all these previous works, we investigate post-processing calibration [22, 23, 33, 50, 61] to adjust the learned model in the testing phase, without modifying the training phase or modeling. Concretely, these methods adjust the predicted confident scores (*i.e.*, the posterior over classes) for each test instance, *e.g.*, by normalizing the classifier norms [22] or by scaling or reducing the logits according to class sizes [23, 33, 61]. Post-processing calibration is quite popular in imbalanced classification but not in long-tailed object detection. To our knowledge, only Li et al. [27] and Tang et al. [49] have studied this approach for object detection[3] . Li et al. [27] applied classifier normalization [22] as a baseline but showed inferior results; Tang et al. [49] developed causal inference calibration rules, which however require a corresponding de-confounded training step. Dave et al. [7] applied methods for calibrating model uncertainty, which are quite different from class-imbalanced learning (see the next paragraph). In this paper, we demonstrate that existing calibration rules for class-imbalanced learning [23, 33, 61] can significantly improve long-tailed object detection, if paired with appropriate ways to deal with the background class and normalized the adjusted logits. We refer the reader to the supplementary material for a comprehensive survey and comparison of the literature.

**Calibration of model uncertainty.** The calibration techniques we employ are different from the ones used for calibrating model uncertainty [11, 24, 25, 34, 38, 63, 64]: we aim to adjust the prediction across classes, while the latter adjusts the predicted probability to reflect the true correctness likelihood. Specifically for long-tailed object detection, Dave et al. [7] applied techniques for calibrating model uncertainty to each object class individually. Namely, a temperature factor or a set of binning grids (*i.e.*, hyper-parameters) has to be estimated for each of the hundreds of classes in the LVIS dataset, leaving the techniques sensitive to hyper-parameter tuning. Indeed, Dave et al. [7] showed that it is quite challenging to estimate those hyper-parameters for tail classes. In contrast, the techniques we apply have only a single hyper-parameter, which can be selected robustly from the training data.

## 3 Post-Processing Calibration for Long-Tailed Object Detection

In this section, we provide the background and notation for long-tailed object detection and instance segmentation, describe our approach **Normalized Calibration** (NORCAL), and discuss its relation to existing post-processing calibration methods.

### 3.1 Background and Notation

Our tasks of interests are object detection and instance segmentation. Object detection focuses on detecting objects via bounding boxes while instance segmentation additionally requires precisely segmenting each object instance in an image. Both tasks involve classifying the object in each box/mask proposal region into one of the pre-defined classes. This classification component is what our proposed approach aims to improve. The most common object classification loss is the cross-entropy (CE) loss,

$$\mathcal{L}_{\text{CE}}(\boldsymbol{x}, \boldsymbol{y}) = -\sum_{c=1}^{C+1} y[c] \times \log\big(p(c|\boldsymbol{x})\big), \tag{1}$$

where $\boldsymbol{y} \in \{0,1\}^{C+1}$ is the one-hot vector of the ground-truth class and $p(c|\boldsymbol{x})$ is the predicted probability (*i.e.*, confidence score) of the proposal $\boldsymbol{x}$ belonging to the class $c$, which is of the form

$$s_c = p(c|\boldsymbol{x}) = \frac{\exp(\phi_c(\boldsymbol{x}))}{\sum_{c'=1}^{C} \exp(\phi_{c'}(\boldsymbol{x})) + \exp(\phi_{C+1}(\boldsymbol{x}))}. \tag{2}$$

---

[3]Calibration in [54, 55] is in the training phase and is not post-processing.

Here, $\phi_c$ is the logit for class $c$, which is usually realized by $\boldsymbol{w}_c^\top f_{\boldsymbol{\theta}}(\boldsymbol{x})$: $\boldsymbol{w}_c$ is the linear classifier associated with class $c$ and $f_{\boldsymbol{\theta}}$ is the feature network. We use $C + 1$ to denote the "background" class.

During testing, a set of "(box/mask proposal, object class, confidence score)" tuples are generated for each image; each proposal can be paired with multiple classes and appears in multiple tuples if the corresponding scores are high enough. The most common evaluation metric for these tuples is average precision (AP), where they are compared against the ground-truths *for each class*[4]. Concretely, the tuples with predicted class $c$ will be gathered, sorted by their scores, and compared with the ground-truths for class $c$. Further, for popular benchmarks such as MSCOCO [28] and LVIS [12], there is a cap $K$ (often set to 300) on the number of detected objects per image, which is enforced usually by including only the tuples with top $K$ confidence scores. Such a cap makes sense in practice, since a scene seldom contains over 300 objects; creating too many, likely noisy tuples can also be annoying to users (*e.g.*, for a camera equipped with object detection).

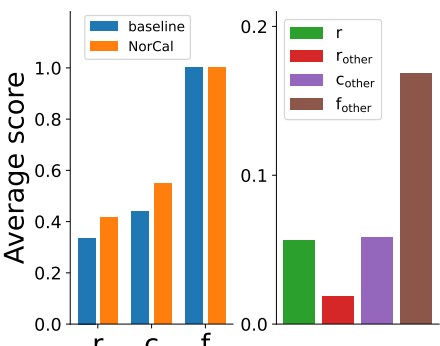

Figure 2: **The effect of long-tailed distributions on LVIS v0.5 [12]. Left:** We show the confidence scores of the top 300 tuples of each image by the baseline Faster R-CNN detector [43] w/o or w/ NORCAL. We average the scores for rare, common, and frequent classes and then linearly scale these averaged scores such that the frequent class has a score of 1. The baseline detector gives frequent objects higher scores, which can be alleviated by NORCAL. **Right:** For tuples of the *rare* class predicted by the baseline Faster R-CNN w/o NORCAL, we further show the average scores of **them** and the average highest scores from another **rare**, **common**, and **frequent** classes on *the same proposals*. The learned baseline detector tends to predict frequent classes for a proposal.

**Long-tailed object detection and instance segmentation: problems and empirical evidence.** Let $N_c$ denote the number of training images of class $c$. A major challenge in long-tailed object detection is that $N_c$ is imbalanced across classes, and the learned classifier using Eq. 1 is biased toward giving higher scores to the head classes (whose $N_c$ is larger) [2, 6, 7, 16]. For instance, in the long-tailed object detection benchmark LVIS [12] whose classes are divided into frequent ($N_c > 100$), common ($100 \geq N_c > 10$), and rare ($N_c \leq 10$), the confidence scores of the rare classes are much smaller than the frequent classes during inference (see Figure 2). As a result, the top $K$ tuples mostly belong to the frequent classes; proposals of the rare classes are often mis-classified as frequent classes, which aligns with the observations by Wang et al. [55].

## 3.2 Normalized Calibration for Long-tailed Object Detection (NORCAL)

Next, we describe the key components of the proposed NORCAL, including confidence score calibration and normalization. The former re-ranks the confidence scores across classes to overcome the bias that rare classes usually have lower confidence scores; the latter helps to re-order the scores of detected tuples within each class for further improving the performance. Both the confidence score calibration and normalization are essential to the success of NORCAL.

**Post-processing calibration and foreground-background decomposition.** We explore applying simple post-calibration techniques from standard multi-way classification [22, 59] to object detection and instance segmentation. The main idea is to scale down the logit of each class $c$ by its size $N_c$ [33, 61]. In our case, however, the background class poses a unique challenge. First, $N_{C+1}$ is ill-defined since nearly all images contain backgrounds. Second, the background regions extracted during model training are drastically different from the foreground object proposals in terms of amounts and appearances. We thus propose to decompose Eq. 2 as follows,

$$p(c|\boldsymbol{x}) = \frac{\sum_{c'=1}^{C} \exp(\phi_{c'}(\boldsymbol{x}))}{\sum_{c'=1}^{C} \exp(\phi_{c'}(\boldsymbol{x})) + \exp(\phi_{C+1}(\boldsymbol{x}))} \times \frac{\exp(\phi_c(\boldsymbol{x}))}{\sum_{c'=1}^{C} \exp(\phi_{c'}(\boldsymbol{x}))}, \qquad (3)$$

---

[4]The difference between AP for object detection and instance segmentation lies in the computation of the intersection over union (IoU): the former based on boxes and the latter based on masks.

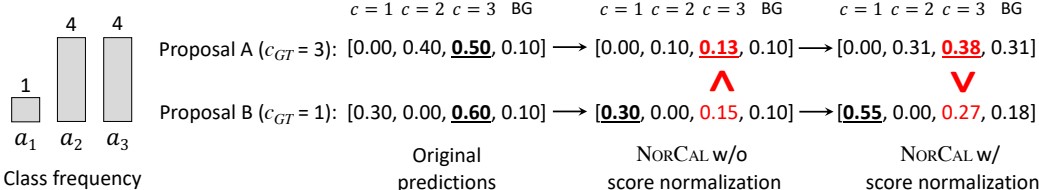

Figure 3: **NORCAL with score normalization can improve AP for head classes.** Here we assume there are three possible foreground classes, and show the ground-truth classes (*i.e.*, $c_{GT}$) and predictions for two object proposals. Bold and underlined numbers indicate the highest scored class for each proposal. The proposed calibration approach and score normalization can be organically coupled together to improve the ranking of personals/tuples for each class. See the text for details.

where the first term on the right-hand side predicts how likely $x$ is foreground (vs. background, *i.e.*, class $C + 1$) and the second term predicts how likely $x$ belongs to class $c$ given that it is foreground. Note that, the background logit $\phi_{C+1}(x)$ only appears in the first term and is compared to all the foreground classes as a whole. In other words, scaling or reducing it does not change the order of confidence scores among the object classes $c \in \{1, \cdots, C\}$. We thus choose to keep $\phi_{C+1}(x)$ intact. Please refer to Section 4 for a detailed analysis, including the effect of adjusting $\phi_{C+1}(x)$.

For the foreground object classes, inspired by Figure 2 and the studies in [7, 33, 61], we propose to scale down the exponential of the logit $\phi_c(x), \forall c \in \{1, \cdots, C\}$, by a positive factor $a_c$,

$$p(c|x) = \frac{\exp(\phi_c(x))/a_c}{\sum_{c'=1}^{C} \exp(\phi_{c'}(x))/a_{c'} + \exp(\phi_{C+1}(x))}, \quad (4)$$

in which $a_c$ should monotonically increase with respect to $N_c$ — such that the scores for head classes will be suppressed. We investigate a simple way to set $a_c$, inspired by [23, 61],

$$a_c = N_c^\gamma, \ \gamma \geq 0, \quad (5)$$

which has a single hyper-parameter $\gamma$ that controls the strength of dependency between $a_c$ and $N_c$. Specifically, if $\gamma = 0$, we recover the original confidence scores in Eq. 2. We investigate other methods beyond Eq. 4 and Eq. 5 in Section 4.

**Hyper-parameter tuning.** Our approach only has a single hyper-parameter $\gamma$ to tune. We observe that we can tune $\gamma$ directly on the training data[5], bypassing the need of a held-out set which can be hard to collect due to the scarcity of examples for the tail classes. Dave et al. [7] also investigate this idea; however, the selected hyper-parameters from training data hurt the test results of rare classes. We attribute this to the fact that their methods have separate hyper-parameters for each class, and that makes them hard to tune.

**The importance of normalization and its effect on AP.** At first glance, our approach NORCAL seems to simply scale down the scores for head classes, and may unavoidably hurt their AP due to the decrease of detected tuples (hence the recall) within the cap. However, we point out that the normalization operation (*i.e.*, sum to 1) in Eq. 4 can indeed improve AP for head classes — *normalization enables re-ordering the scores of tuples within each class.*

Let us consider a three-class example (see Figure 3), in which $c = 1$ is a tail class, $c = 2$ and $c = 3$ are head classes, and $c = 4$ is the background class. Suppose two proposals are found from an image: proposal $A$ has scores $[0.0, 0.4, 0.5, 0.1]$ and the true label $c_{GT} = 3$; proposal $B$ has scores $[0.3, 0.0, 0.6, 0.1]$ and the true label $c_{GT} = 1$. Before calibration, proposal $B$ is ranked higher than $A$ for $c = 3$, resulting in a low AP. Let us assume $a_1 = 1$ and $a_2 = a_3 = 4$. If we simply divide the scores of object classes by these factors, proposal $B$ will still be ranked higher than $A$ for $c = 3$. However, by applying Eq. 4, we get the new scores for proposal $A$ as $[0.0, 0.31, 0.38, 0.31]$ and for proposal $B$ as $[0.55, 0.0, 0.27, 0.18]$ — proposal $A$ is now ranked higher than $B$ for $c = 3$, leading to a higher AP for this class. As will be seen in Section 4, such a "re-ranking" property is the key to making NORCAL excel in AP for all classes as well as in other metrics like AP$^{\text{Fixed}}$ [7].

---

[5]Unlike imbalanced classification in which the learned classifier ultimately achieves $\sim 100\%$ accuracy on the training data [61, 62] (so hyper-parameter tuning using the training data becomes infeasible), a long-tailed object detector can hardly achieve $100\%$ AP per class even on the training data.

### 3.3 Comparison to Existing Work

Li et al. [27] investigated classifier normalization [22] for post-processing calibration. They modified the calculation of $\phi_c$ from $\boldsymbol{w}_c^\top f_{\boldsymbol{\theta}}(\boldsymbol{x})$ to $\frac{\boldsymbol{w}_c^\top}{\|\boldsymbol{w}_c\|_2^\gamma} f_{\boldsymbol{\theta}}(\boldsymbol{x})$, building upon the observation that the classifier weights of head classes tend to exhibit larger norms [22]. The results, however, were much worse than their proposed cost-sensitive method BaGS. They attributed the inferior result to the background class, and had combined two models, with or without classifier normalization, attempting to improve the accuracy. Our decomposition in Eq. 3 suggests a more straightforward way to handle the background class. Moreover, $N_c$ provides a better signal for calibration than $\|\boldsymbol{w}_c\|_2$, according to [23, 61]. We provide more discussions and comparison results in the supplementary material.

### 3.4 Extension to Multiple Binary Sigmoid Classifiers

Many existing models for long-tailed object detection and instance segmentation are based on multiple binary classifiers instead of the softmax classifier [19, 42, 46, 48, 56]. That is, $s_c$ in Eq. 2 becomes

$$s_c = \frac{1}{1 + \exp(-\boldsymbol{w}_c^\top f_{\boldsymbol{\theta}}(\boldsymbol{x}))} = \frac{1}{1 + \exp(-\phi_c(\boldsymbol{x}))} = \frac{\exp(\phi_c(\boldsymbol{x}))}{\exp(\phi_c(\boldsymbol{x})) + 1}, \qquad (6)$$

in which $\boldsymbol{w}_c$ treats every class $c' \neq c$ and the background class together as the "negative" class. In other words, the background logit $\phi_{C+1} = \boldsymbol{w}_{C+1}^\top f_{\boldsymbol{\theta}}(\boldsymbol{x})$ in Eq. 2 is not explicitly learned.

Our post-processing calibration approach can be extended to multiple binary classifiers as well. For example, Eq. 4 becomes

$$s_c = \frac{\exp(\phi_c(\boldsymbol{x}))/a_c}{\exp(\phi_c(\boldsymbol{x}))/a_c + 1}. \qquad (7)$$

We note that solely calibrating the scores can re-rank the detected tuples across classes within each image such that rare and common objects, which initially have lower scores, could be included in the cap to largely increase the recall. Therefore, as will be shown in the experimental results, the improvement for multiple binary classifiers mainly comes from the rare and common objects.

However, one drawback of the score calibration alone is the infeasibility of normalization across classes; $s_c$ does not necessarily sum to 1, making it hard to re-order the scores of tuples within each class. Forcing the confidence scores across classes of each proposal to sum to 1 would inevitably turn many background patches into foreground proposals due to the lack of the background logit $\phi_{C+1}$.

## 4 Experiments

### 4.1 Setup

**Dataset.** We validate NORCAL on the LVIS v1 dataset [12], a benchmark dataset for large-vocabulary instance segmentation which has 100K/19.8K/19.8K training/validation/test images. There are 1,203 categories, divided into three groups based on the number of training images per class: rare (1–10 images), common (11–100 images), and frequent (>100 images). *All results are reported on the validation set of LVIS v1.* For comparisons to more existing works and different tasks, we also conduct detailed experiments and analyses on LVIS v0.5 [12], Objects365 [44], MSCOCO [28], and image classification datasets in the supplementary material.

**Evaluation metrics.** We adopt the standard mean **Average Precision (AP)** [28] for evaluation. The cap over detected objects per image is set as 300 (cf. Section 3.1). Following [12], we denote the mean AP for rare, common, and frequent categories by $AP_r$, $AP_c$, and $AP_f$, respectively. We also report results with a complementary metric **$AP^{Fixed}$** [7], which replaces the cap over detected objects per image by a cap over detected objects per class from the entire validation set. Namely, $AP^{Fixed}$ removes the competition of confidence scores among classes within an image, making itself category-independent. We follow [7] to set the per-class cap as $10,000$. Instead of Mask AP, we also report the results in **$AP^{Fixed}$** [7] with **Boundary IoU**, following the standard evaluation metric in LVIS Challenge 2021[6]. Meanwhile, we report **$AP^b$**, which assesses the AP for the bounding boxes produced by the instance segmentation models.

---

[6] https://www.lvisdataset.org/challenge_2021.

Table 1: **Comparison of instance segmentation on the validation set of LVIS v1.** NORCAL provides solid improvement to existing models. †: with EQL, we see a slight drop on the frequent classes due to the infeasibility of score normalization across classes with multiple binary classifiers. ⋆: models from [66]. ‡: models from [48].

| Backbone | Method | NORCAL | AP | $AP_r$ | $AP_c$ | $AP_f$ | $AP^b$ |
|---|---|---|---|---|---|---|---|
| R-50 [17] | EQL [46]‡ | ✗ | 18.60 | 2.10 | 17.40 | 27.20 | 19.30 |
| | | ✓ | (+2.30) 20.90 | (+3.90) 6.00 | (+3.80) 21.20 | †(-0.10) 27.10 | (+2.50) 21.80 |
| | cRT [22]‡ | ✗ | 22.10 | 11.90 | 20.20 | 29.00 | 22.20 |
| | | ✓ | (+2.20) 24.30 | (+3.50) 15.40 | (+2.70) 22.90 | (+0.70) 29.70 | (+1.50) 23.70 |
| | RFS [12]⋆ | ✗ | 22.58 | 12.30 | 21.28 | 28.55 | 23.25 |
| | | ✓ | (+2.65) 25.22 | (+7.03) 19.33 | (+2.88) 24.16 | (+0.43) 28.98 | (+2.83) 26.08 |
| | MosaicOS [66] | ✗ | 24.45 | 18.17 | 22.99 | 28.83 | 25.05 |
| | | ✓ | (+2.32) 26.76 | (+5.69) 23.86 | (+2.82) 25.82 | (+0.27) 29.10 | (+2.73) 27.77 |
| R-101 [17] | RFS [12]⋆ | ✗ | 24.82 | 15.18 | 23.71 | 30.31 | 25.45 |
| | | ✓ | (+2.43) 27.25 | (+5.61) 20.79 | (+2.74) 26.45 | (+0.68) 30.99 | (+2.60) 28.05 |
| | MosaicOS [66] | ✗ | 26.73 | 20.53 | 25.78 | 30.53 | 27.41 |
| | | ✓ | (+2.30) 29.03 | (+5.85) 26.38 | (+2.37) 28.15 | (+0.66) 31.19 | (+2.55) 29.96 |
| X-101 [60] | RFS [12]⋆ | ✗ | 26.67 | 17.60 | 25.58 | 31.89 | 27.35 |
| | | ✓ | (+1.25) 27.92 | (+2.15) 19.75 | (+1.61) 27.19 | (+0.45) 32.34 | (+1.49) 28.83 |
| | MosaicOS [66] | ✗ | 28.29 | 21.75 | 27.22 | 32.35 | 28.85 |
| | | ✓ | (+1.52) 29.81 | (+3.97) 25.72 | (+1.70) 28.92 | (+0.24) 32.59 | (+1.71) 30.56 |

**Implementation details and variants.** We apply NORCAL to post-calibrate several representative baseline models, for which we use the released checkpoints from the corresponding papers. We focus on models that have *a softmax classifier or multiple binary classifiers* for assigning labels to proposals[7]. For NORCAL, (a) we investigate different mechanisms by applying post-calibration to the classifier logits, exponentials, or probabilities (cf. Eq. 4); (b) we study different types of calibration factor $a_c$, using the class-dependent temperature (CDT) [61] presented in Eq. 5 or the effective number of samples (ENS) [6]; (c) we compare with or without score normalization. We tune the only hyper-parameter of NORCAL (*i.e.*, in $a_c$) on training data.

## 4.2 Main Results

**NORCAL effectively improves baselines in diverse scenarios.** We first apply NORCAL to representative baselines for instance segmentation: (1) Mask R-CNN [18] with feature pyramid networks [29], which is trained with repeated factor sampling (RFS), following the standard training procedure in [12]; (2) re-sampling/cost-sensitive based methods that have a multi-class classifier, *e.g.*, cRT [22]; (3) re-sampling/cost-sensitive based methods that have multiple binary classifiers, *e.g.*, EQL [46]; (4) data augmentation based methods, *e.g.*, a state-of-the-art method MosaicOS [66]. *Please see the supplementary material for a comparison with other existing methods.*

Table 1 provides our main results on LVIS v1. NORCAL achieves consistent gains on top of all the models of different backbone architectures. For instance, for RFS [12] with ResNet-50, the overall AP improves from 22.58% to 25.22%, including ∼ 7%/3% gains on $AP_r$/$AP_c$ for rare/common objects. Importantly, we note that NORCAL's improvement is on almost all the evaluation metrics (columns), demonstrating a key strength of NORCAL that is not commonly seen in literature: achieving overall gains without sacrificing the $AP_f$ on frequent classes. We attribute this to the score normalization operation of NORCAL: unlike [7] which only re-ranks scores across categories, NORCAL further re-ranks the scores within each category. Indeed, the only performance drop in Table 1 is on frequent classes for EQL, which is equipped with multiple binary classifiers such that score normalization across classes is infeasible (cf. Section 3.4). We provide more discussions in the ablation studies.

**Comparison to existing post-calibration methods.** We then compare our NORCAL to other post-calibration techniques. Specifically, we compare to those in [7] on the LVIS v1 instance segmentation task, including Histogram Binning [63], Bayesian binning into quantiles (BBQ) [34], Beta calibration [24], isotonic regression [64], and Platt scaling [38]. We also compare to classifier normalization ($\tau$-normalized) [22, 27] on the LVIS v0.5 object detection task. All the hyper-parameters for calibration are tuned from the training data.

---

[7]Several existing methods (*e.g.*, [27, 48, 53]) develop specific classification rules to which NORCAL cannot be directly applied.

Table 2: **Comparison to other existing post-calibration methods.** NORCAL outperforms methods studied in [7] and [27]. †: w/o RFS [12].

| Segmentation on v1 | AP | $AP_r$ | $AP_c$ | $AP_f$ |
|---|---|---|---|---|
| RFS [12] | 22.58 | 12.30 | 21.28 | 28.55 |
| w/ HistBin [63] | 21.82 | 11.28 | 20.31 | 28.13 |
| w/ BBQ (AIC) [34] | 22.05 | 11.41 | 20.72 | 28.21 |
| w/ Beta calibration [24] | 22.55 | 12.29 | 21.27 | 28.49 |
| w/ Isotonic reg. [64] | 22.43 | 12.19 | 21.12 | 28.41 |
| w/ Platt scaling [38] | 22.55 | 12.29 | 21.27 | 28.49 |
| w/ NORCAL | 25.22 | 19.33 | 24.16 | 28.98 |

| Detection on v0.5 | $AP^b$ | $AP_r^b$ | $AP_c^b$ | $AP_f^b$ |
|---|---|---|---|---|
| Faster R-CNN [43]† | 20.98 | 4.13 | 19.70 | 29.30 |
| w/ $\tau$-normalized [27]† | 21.61 | 6.18 | 20.99 | 28.54 |
| w/ NORCAL † | 23.87 | 6.98 | 24.17 | 30.24 |

Table 3: **Ablation studies of** NORCAL **with various modeling choices and mechanisms.** We report results on LVIS v1 instance segmentation. CAL: calibration mechanism. NOR: class score normalization. The best ones are in bold.

| $a_c$ | CAL | NOR | AP | $AP_r$ | $AP_c$ | $AP_f$ |
|---|---|---|---|---|---|---|
| Baseline | $\exp(\phi_c(\boldsymbol{x}))$ | ✓ | 22.58 | 12.30 | 21.28 | 28.55 |
| $\dfrac{1-\gamma^{N_c}}{1-\gamma}$ (ENS [6]) | $\exp(\phi_c(\boldsymbol{x})/a_c)$ | ✗ | 23.66 | 14.55 | 22.36 | **29.11** |
| | | ✓ | 23.96 | 15.84 | 22.61 | 29.04 |
| | $p(c|\boldsymbol{x})/a_c$ | ✗ | 24.18 | 18.88 | 22.95 | 27.87 |
| | | ✓ | 24.85 | **19.43** | 23.67 | 28.54 |
| | $\exp(\phi_c(\boldsymbol{x}))/a_c$ | ✗ | 17.49 | 14.16 | 17.20 | 19.27 |
| | | ✓ | 24.85 | **19.43** | 23.67 | 28.54 |
| $N_c^{\gamma}$ (CDT [61]) | $\exp(\phi_c(\boldsymbol{x})/a_c)$ | ✗ | 17.52 | 14.04 | 17.38 | 19.20 |
| | | ✓ | 24.77 | 17.99 | 23.81 | 28.83 |
| | $p(c|\boldsymbol{x})/a_c$ | ✗ | 24.50 | 18.34 | 23.42 | 28.41 |
| | | ✓ | **25.22** | 19.33 | **24.16** | 28.98 |
| | $\exp(\phi_c(\boldsymbol{x}))/a_c$ | ✗ | 17.52 | 13.93 | 17.24 | 19.41 |
| | | ✓ | **25.22** | 19.33 | **24.16** | 28.98 |

Table 2 shows the results. NORCAL significantly outperforms other techniques on both tasks and can improve AP for all the classes. We attribute the improvement over methods studied in [7] to two reasons: first, NORCAL has only one hyper-parameter, while calibration methods in [7] have hyper-parameters for every category and thus are sensitive to tune; second, NORCAL performs score normalization, while [7] does not. Compared to [22, 27], the use of per-class data count in NORCAL has been shown to outperform classifier norms for calibrating classifiers [23, 61].

## 4.3 Ablation Studies and Analysis

We mainly conduct the ablation studies on the Mask R-CNN model [18] (with ResNet-50 backbone [17] and feature pyramid networks [29]), trained with repeated factor sampling (RFS) [12].

**Effect of calibration mechanisms.** In addition to reducing the logits, *i.e.*, scaling down their exponentials (*i.e.*, $\exp(\phi_c(\boldsymbol{x}))/a_c$ in Eq. 4), we investigate another two ways of score calibration. Specifically, we scale down the output logits from the network (*i.e.*, $\phi_c(\boldsymbol{x})/a_c$) or the probabilities from the classifier (*i.e.*, $p(c|\boldsymbol{x})/a_c$). Again, we keep the background class intact and apply score normalization. In Table 3, we see that scaling down the exponentials and probabilities perform the same[8] and outperform scaling down logits. We note that, logits can be negative; thus, scaling them down might instead increases the scores. In contrast, exponentials and probabilities are non-negative, scaling them down thus are guaranteed to reduce the scores of frequent classes more than rare classes.

**Effect of calibration factors** $a_c$**.** Beyond the class-dependent temperature (CDT) [61] presented in Eq. 5, we study an alternative factor, inspired by the effective number of samples (ENS) [6]. Specifically, we study $a_c = (1 - \gamma^{N_c})/(1 - \gamma)$ with $\gamma \in [0, 1)$. Same as CDT, ENS has a single hyper-parameter $\gamma$ that controls the degree of dependency between $a_c$ and $N_c$. If $\gamma = 0$, we recover the original confidence scores. We report the comparison of these two calibration factors in Table 3. With appropriate post-calibration mechanisms, both provide consistent gains over the baseline model.

**Importance of score normalization.** Again in Table 3, we compare NORCAL with or without score normalization across classes. That is, whether we include the denominator in Eq. 4 or not. By applying normalization, we see that NORCAL can improve all categories, including frequent objects. Moreover, it is applicable to different types of calibration mechanisms as well as calibration factors. In contrast, the results without normalization degrade at frequent classes and sometimes even at common and rare classes. We attribute this to two reasons: first, score normalization enables the detected tuples of each class to be re-ranked (cf. Figure 3); second, with the background logits in the denominator, the calibrated and normalized scores can effectively prevent background patches from being classified

---

[8]With class score normalization, they are mathematically the same.

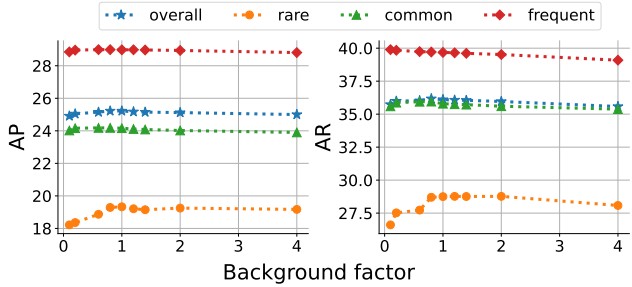 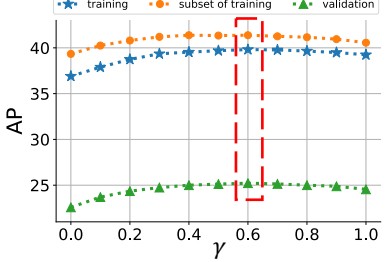

Figure 4: **Results of precision and recall by adjusting background class scores.** Results are on v1 instance segmentation.

Figure 5: **Calibration factor $\gamma$ can be robustly tuned using training data.**

Table 4: **NORCAL improves average precision and recall.** Results are on LVIS v1 instance segmentation.

| (a) Average Precision (AP) | | | | | (b) Average Recall (AR) | | | | |
|---|---|---|---|---|---|---|---|---|---|
| | AP | $AP_r$ | $AP_c$ | $AP_f$ | | AR | $AR_r$ | $AR_c$ | $AR_f$ |
| RFS [12] | 22.58 | 12.30 | 21.28 | 28.55 | RFS [12] | 30.61 | 13.73 | 28.64 | 40.24 |
| w/ NORCAL | 25.22 | 19.33 | 24.16 | 28.98 | w/ NORCAL | 36.10 | 28.75 | 35.79 | 39.68 |

Table 5: **NORCAL can improve the baseline model in AP$^{Fixed}$ and AP$^{Fixed}$ with Boundary IoU.** The baseline model uses ResNet-50 as the backbone with RFS [12]. Results are reported on LVIS v1 instance segmentation.

| (a) AP Fixed | | | | | (b) AP Fixed with Boundary IoU | | | | |
|---|---|---|---|---|---|---|---|---|---|
| | AP | $AP_r$ | $AP_c$ | $AP_f$ | | AP | $AP_r$ | $AP_c$ | $AP_f$ |
| RFS [12] | 25.68 | 20.07 | 24.82 | 29.11 | RFS [12] | 19.88 | 14.76 | 19.32 | 22.76 |
| w/ NORCAL | 26.26 | 20.56 | 25.39 | 29.73 | w/ NORCAL | 20.25 | 14.99 | 19.77 | 23.10 |

into foreground objects. Please be referred to the supplementary material for additional results and ablation studies on sigmoid-based detectors (*i.e.*, BALMS [42] and RetinaNet [30]).

**How to handle the background class?** NORCAL does not calibrate the background class logit. We ablate this design by *multiplying* the exponential of the background logit with a background calibration factor $\beta$, *i.e.*, $\exp(\phi_{C+1}(\boldsymbol{x})) \times \beta$. If $\beta = 1$, there is no calibration on background class. Figure 4 shows the average precision and recall of the model with NORCAL w.r.t different $\beta$. We see consistent performance for $\beta \geq 1$. For $\beta < 1$, the average precision drops along with reduced $\beta$, especially for the rare classes whose average recall also drops. We note that, in the extreme case with $\beta = 0$, the background class will not contribute to the final calibrated score. Thus, many background patches may be classified as foregrounds and ranked higher than rare proposals. These results and explanation justifies one key ingredient of NORCAL— keeping the background logit intact.

**Sensitivity to the calibration factor.** NORCAL has one hyper-parameter: $\gamma$ in the calibration factor $a_c$, which controls the strength of calibration. We find that this can be tuned robustly on the *training* data, even on a 5K subset of *training* images: as shown in Figure 5, the AP trends on the training and validation sets at different $\gamma$ are close to each other. In our experiments, we find that this observation applies to different models and backbone architectures.

**NORCAL reduces false positives and re-ranks predictions within each class.** In Table 4, we show that NORCAL can improve the AR for all classes but frequent objects (with a slight drop). The gains on AP for frequent classes thus suggest that NORCAL can re-rank the detected tuples within each class, pushing many false positives to have scores lower than true positives.

**NORCAL is effective in AP$^{Fixed}$ [7] and Boundary IoU [5].** Table 5 reports the results in AP$^{Fixed}$ and AP$^{Fixed}$ with Boundary IoU. We see that NORCAL is metric-agnostic and can consistently improve the baseline model in all groups of categories. It suggests that the improvements are due to both across-class and within-class re-ranking.

**Limiting detections per image.** Finally, we evaluate NORCAL by changing the cap on the number of detections per image. Specifically, we investigate reducing the default number of 300. The rationale is that an image seldom contains over 300 objects. Indeed, each LVIS [12] image is annotated with around 12 object instances on average. We note that, to perform well in a smaller cap requires a

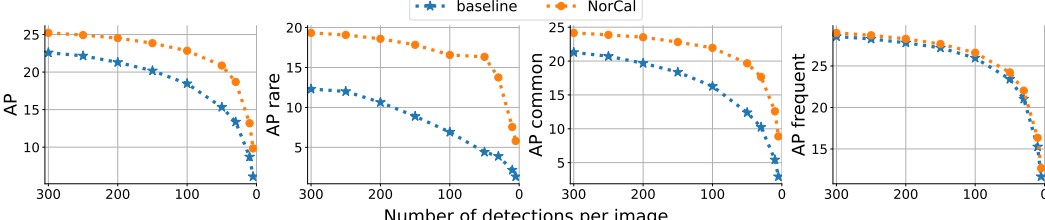

Figure 6: **Limits on the number of detections per image.** To perform well in a small cap, a model must rank true positives higher such that they can be included in the cap. NORCAL performs much better than the baseline.

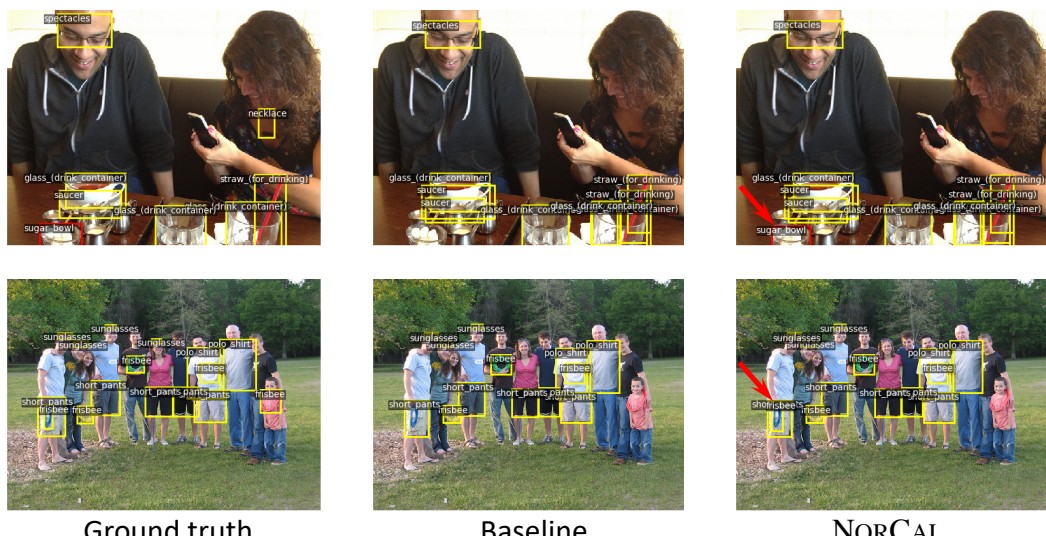

Ground truth       Baseline       NORCAL

Figure 7: **Qualitative results.** We superimpose red arrows to show the improvement, and Yellow and red boxes to indicate the ground truth labels of frequent and rare classes. In the first example, NORCAL successfully detects the rare object *sugar bowl* without sacrificing other predictions. In the second example, even surprisingly, it can detect a missed frequent object *frisbee* by the baseline.

model to rank most true positives in the front such that they can be included in the cap. In Figure 6, NORCAL shows superior performance against the baseline model under all settings. It is worth noting that NORCAL achieves better performance even using a strict 100 detections per image than the baseline model with 300.

**Qualitative results.** We show qualitative bounding box results on LVIS v1 in Figure 7. We compare the ground truths, the results of the baseline, and the results of NORCAL. NORCAL can not only detect more objects from the rare categories that may be overlooked by the baseline detector, but also improve the detection results on frequent objects. For instance, in the upper example of Figure 7, NORCAL discovers a rare object "sugar bowl" without sacrificing any other frequent objects. Moreover, NORCAL can improve the frequent classes, as shown in the bottom example of Figure 7. Please see the supplementary material for more qualitative results.

## 5 Conclusion

We present a post-processing calibration method called NORCAL for addressing long-tailed object detection and instance segmentation. Our method is simple yet effective, requires no re-training of the already trained models, and can be compatible with many existing models to further boost the state of the art. We conduct extensive experiments to demonstrate the effectiveness of our method in diverse settings, as well as to validate our design choices and analyze our method's mechanisms. We hope that our results and insights can encourage more future works on exploring the power of post-processing calibration in long-tailed object detection and instance segmentation.

## Acknowledgments and Funding Transparency Statement

This research is partially supported by NSF IIS-2107077 and the OSU GI Development funds. We are thankful for the generous support of the computational resources by the Ohio Supercomputer Center. We thank Zhiyun Lu (Google) for feedback on an early draft of this paper and Han-Jia Ye (Nanjing University) for the help on image classification experiments.

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
