# On Model Calibration for Long-Tailed Object Detection and Instance Segmentation

**Tai-Yu Pan**[1*]   **Cheng Zhang**[1*]   **Yandong Li**[2]   **Hexiang Hu**[2]
**Dong Xuan**[1]   **Soravit Changpinyo**[2]   **Boqing Gong**[2]   **Wei-Lun Chao**[1]

[1]The Ohio State University    [2]Google Research

## Supplementary Material

In this supplementary material, we provide details and additional results omitted in the main texts.

## A   Additional Discussion on Related Work

### A.1   Long-Tailed Object Detection and Instance Segmentation

Existing works can be categorized into re-sampling, cost-sensitive learning, and data augmentation.

**Re-sampling** changes the training data distribution — by sampling rare class data more often than frequent class ones — to mitigate the long-tailed distribution. Re-sampling is widely adopted as a simple but effective baseline approach [1, 8, 27]. For example, repeat factor sampling (RFS) [8] sets a repeat factor (*i.e.*, sampling frequency) for each image based on the rarest object within that image; class-aware sampling [27] samples a uniform amount of images per class for each mini-batch. Since an image can contain multiple object classes, Chang et al. [1] proposed to re-sample on both the image and object instance levels. RFS is the baseline approach used for the LVIS dataset [8].

**Cost-sensitive learning** is the most popular category, which adjusts the cost of mis-classifying an instance or the loss of learning from an instance according to its true class label. Re-weighting is the simplest method of this kind, which gives each instance a class-specific weight in calculating the total loss (usually, tail classes with larger weights). The equalization loss (EQL) [28] and EQL v2 [29] ignore the negative gradients for rare class classifiers or equalize the positive-negative gradient ratio for each class to balance the training, respectively. The drop loss [11] improves EQL by specifically handling the background class via re-weighting.  The seesaw loss [32] proposes a re-weighting

---

*Equal contributions

scheme by combining the dataset statistics and training dynamics. Forest R-CNN [36] leverages the class hierarchical for knowledge transfer and introduces new losses for hierarchical classification.

Instead of applying the new loss functions during the entire training phase, several recent methods decouple the training phase into two stages [12, 13, 16, 24, 33–35, 43]. At the first stage, the object detector is trained normally just like on a relatively balanced dataset such as MSCOCO [17]. Then in the second stage, re-sampling or cost-sensitive learning is employed, usually for re-training or fine-tuning only the classification network. Such a pipeline is shown to learn both better features and classifier. For example, two-stage fine-tuning approach (TFA) [35] first trains a base detector using only common and frequent classes, and then fine-tune the classifier and box regressor with re-sampling. Similar ideas are adopted in classifier re-training (cRT) [13], SimCal [33], balanced softmax (BSM) [24], balanced group softmax (BaGS) [16], DisAlign [43], and ACSL [34], which develop strategies or losses to re-train the classifier. Learning to segment the tail (LST) [12] takes an incremental learning approach to gradually learn from the head to tail classes in multiple stages.

**Data augmentation** improves long-tailed object detection by augmenting data for the tail classes. DLWL [23] and MosaicOS [42] leveraged weakly-supervised data from YFCC-100M [30], ImageNet [5], and Internet to augment the long-tailed LVIS dataset [8]. Copy-Paste [6] self-augments the LVIS dataset by copying object instances from one image and paste to the others. Instead of augmenting images, FASA [41] generates class-wise virtual features using a Gaussian prior whose parameters are estimated from features of real data.

### A.2  Calibration of Model Uncertainty

We note that, the calibration rules we apply are different from the ones used for calibrating model uncertainty [7]: we aim to adjust the prediction across classes, while the latter adjusts the predicted probability to reflect the true correctness likelihood. For calibrating model uncertainty, representative methods are Platt scaling [22], histogram binning [39], Bayesian binning into quantiles (BBQ) [21], isotonic regression [40], temperature scaling [7], beta and Dirichlet calibration [14, 15], etc.

## B  Experimental Setups

### B.1  Baseline Methods

Our approach NORCAL is model-agnostic as long as the detector has ***a softmax classifier or multiple binary sigmoid classifiers for the objects and the background***. Thus, we focus on those methods as long as the pre-trained models are applicable and public:

- The baseline Mask R-CNN [10] model with feature pyramid networks [18], which is trained with repeated factor sampling (RFS), following the standard training procedure in [8].
- Re-sampling/cost-sensitive based methods that have a multi-class classifier for the foreground objects and the background class, *e.g.*, cRT [13] and TFA [35].
- Re-sampling/cost-sensitive based methods that have multiple binary sigmoid-based classifiers, *e.g.*, EQL [28], BALMS [24], and RetinaNet with focal loss [19].
- Data augmentation based methods, *e.g.*, MosaicOS [42]. MosaicOS augments LVIS with images from ImageNet [5], which can improve the feature network of an object detector like Faster R-CNN [25] or Mask R-CNN [9].

We note that, several methods change the decision/classification rules. For example, EQL v2 [29] and Seesaw [32] adopt a separate background or objectness branch during the training and inference. Some other methods (BaGS [16] and Forest R-CNN [36]) re-organize the category groups and apply either a group-based softmax classifier or hierarchical classification. Therefore, it is not immediately obvious how to apply calibration to them.

### B.2  Implementation

NORCAL is easy to implement and requires no re-training of the model. We follow Eq. 4 and Eq. 5 of the main paper to apply NORCAL to the existing models. For all the baseline detectors, we directly take the released models from the corresponding papers without any modifications. We report the

Table A: **Instance segmentation results on the validation set of LVIS v1.** Our method NORCAL can improve all baseline models with different backbones to which it is applied. Seesaw [32] applies a stronger $2\times$ training schedule while other methods are with $1\times$ schedule. †: slight performance drop on sigmoid-based detectors. ⋆: models from [42]. ‡: models from [29]. ♣: results from [6].

| Backbone | Method | NORCAL | AP | $AP_r$ | $AP_c$ | $AP_f$ | $AP^b$ |
|---|---|---|---|---|---|---|---|
| R-50 | DropLoss [11] | | 19.80 | 3.50 | 20.00 | 26.70 | 20.40 |
| | BaGS [16] | | 23.10 | 13.10 | 22.50 | 28.20 | 25.76 |
| | Forest R-CNN [36] | | 23.20 | 14.20 | 22.70 | 27.70 | 24.60 |
| | RIO [1] | | 23.70 | 15.20 | 22.50 | 28.80 | 24.10 |
| | EQL v2 [29] | | 23.70 | 14.90 | 22.80 | 28.60 | 24.20 |
| | DisAlign [43] | | 24.30 | 8.50 | **26.30** | 28.10 | 23.90 |
| | Seesaw [32]$^{2\times}$ | | 25.40 | 15.90 | 24.70 | **30.40** | 25.60 |
| | Seesaw w/ RFS [32]$^{2\times}$ | | 26.40 | 19.60 | 26.10 | 29.80 | 27.40 |
| | EQL [28]‡ | | 18.60 | 2.10 | 17.40 | 27.20 | 19.30 |
| | | ✓ | (+2.30) 20.90 | (+3.90) 6.00 | (+3.80) 21.20 | †(-0.10) 27.10 | (+2.50) 21.80 |
| | cRT [13]‡ | | 22.10 | 11.90 | 20.20 | 29.00 | 22.20 |
| | | ✓ | (+2.20) 24.30 | (+3.50) 15.40 | (+2.70) 22.90 | (+0.70) 29.70 | (+1.50) 23.70 |
| | RFS [8]⋆ | | 22.58 | 12.30 | 21.28 | 28.55 | 23.25 |
| | | ✓ | (+2.65) 25.22 | (+7.03) 19.33 | (+2.88) 24.16 | (+0.43) 28.98 | (+2.83) 26.08 |
| | MosaicOS [42] | | 24.45 | 18.17 | 22.99 | 28.83 | 25.05 |
| | | ✓ | (+2.32) **26.76** | (+5.69) **23.86** | (+2.82) 25.82 | (+0.27) 29.10 | (+2.73) **27.77** |
| R-101 | Seesaw [32]$^{2\times}$ | | 27.10 | 18.70 | 26.30 | 31.70 | 27.40 |
| | Seesaw w/ RFS [32]$^{2\times}$ | | 28.10 | 20.00 | 28.00 | **31.90** | 28.90 |
| | RFS [8]⋆ | | 24.82 | 15.18 | 23.71 | 30.31 | 25.45 |
| | | ✓ | (+2.43) 27.25 | (+5.61) 20.79 | (+2.74) 26.45 | (+0.68) 30.99 | (+2.60) 28.05 |
| | MosaicOS [42] | | 26.73 | 20.53 | 25.78 | 30.53 | 27.41 |
| | | ✓ | (+2.30) **29.03** | (+5.85) **26.38** | (+2.37) **28.15** | (+0.66) 31.19 | (+2.55) **29.96** |
| X-101 | cRT [13]♣ | | 27.20 | 19.60 | 26.00 | 31.90 | – |
| | RIO [1] | | 27.50 | 18.80 | 26.70 | 32.30 | 28.50 |
| | RFS [8]⋆ | | 26.67 | 17.60 | 25.58 | 31.89 | 27.35 |
| | | ✓ | (+1.25) 27.92 | (+2.15) 19.75 | (+1.61) 27.19 | (+0.45) 32.34 | (+1.49) 28.83 |
| | MosaicOS [42] | | 28.29 | 21.75 | 27.22 | 32.35 | 28.85 |
| | | ✓ | (+1.52) **29.81** | (+3.97) **25.72** | (+1.70) **28.92** | (+0.24) **32.59** | (+1.71) **30.56** |

results on the validation set with the best hyper-parameter tuned on training images for all models and benchmarks. The implementations are mainly based on the Detectron2 [37] or MMdetection [2] framework. We run our experiments on 4 NVIDIA RTX A6000 GPUs with AMD 3960X CPUs.

### B.3  Inference and Evaluation

We follow the standard evaluation protocol for the LVIS benchmark [8]. Specifically, during the inference, the threshold of confidence score is set to $10^{-4}$, and we keep the top 300 proposals as the predicted results. No test time augmentation is used. We adopt the standard mean Average Precision (AP) and denote the AP for rare, common, and frequent categories by $AP_r$, $AP_c$, and $AP_f$, respectively. For the object detection results on LVIS v0.5, we report the box AP for each category.

## C  Additional Experimental Results and Analyses

Due to space limitations, we only reported the results of NORCAL with strong baseline models in the main paper (cf. Table 1). In this section, we provide detailed comparisons with more existing works on LVIS [8] v1 and v0.5. We also examine NORCAL on MSCOCO dataset [17]. Moreover, we conduct further analyses and ablation studies of our method.

### C.1  Results on LVIS v1 Instance Segmentation

We summarize the results of instance segmentation on LVIS v1 in Table A. As mentioned in Section B.1, several methods (*e.g.*, BaGS [16], EQL v2 [29], Seesaw [32]) change the decision/classification rules and it is not immediately obvious how to apply calibration to them. Nevertheless, we include their results for comparison. We observe, for example, that NORCAL can improve

Table B: **Instance segmentation results on the validation set of LVIS v0.5.** Our method NORCAL can improve a simple baseline such as RFS [8] to match or outperform all methods with different backbone models. †: slight performance drop on sigmoid-based detectors. ⋆: models from Detectron2 [37]. ‡: models from [24] (the results are slightly different from those reported in [24]).

| Backbone | Method | NORCAL | AP | $AP_r$ | $AP_c$ | $AP_f$ | $AP^b$ |
|---|---|---|---|---|---|---|---|
| R-50 | EQL [28] | | 22.80 | 11.30 | 24.70 | 25.10 | 23.30 |
| | LST [12] | | 23.00 | – | – | – | – |
| | SimCal [33] | | 23.40 | 16.40 | 22.50 | 27.20 | – |
| | DropLoss [11] | | 25.50 | 13.20 | 27.90 | 27.30 | 25.10 |
| | Forest R-CNN [36] | | 25.60 | 18.30 | 26.40 | 27.60 | 25.90 |
| | BaGS [16] | | 26.25 | 17.97 | 26.91 | 28.74 | 25.76 |
| | DisAlign [43] | | 24.20 | 8.50 | 26.20 | 28.00 | 23.90 |
| | RIO [1] | | 26.00 | 18.90 | 26.20 | 28.50 | – |
| | EQL v2 [29] | | 27.10 | 18.60 | 27.60 | **29.90** | 27.00 |
| | BALMS [24]‡ | | 26.97 | 17.31 | 28.07 | 29.47 | 26.42 |
| | | ✓ | (+0.55) 27.52 | (+2.02) 19.33 | (+0.75) 28.82 | †(-0.30) 29.17 | (+0.38) 26.80 |
| | RFS [8]⋆ | | 24.39 | 15.98 | 23.97 | 28.26 | 23.64 |
| | | ✓ | (+2.23) 26.61 | (+2.73) 18.71 | (+3.40) 27.37 | (+0.57) 28.83 | (+2.36) 26.00 |
| | MosaicOS [42] | | 26.28 | 19.65 | 26.62 | 28.49 | 25.76 |
| | | ✓ | (+1.69) **27.97** | (+3.57) **23.22** | (+2.02) 28.64 | (+0.54) 29.03 | (+1.86) **27.61** |
| R-101 | EQL [28] | | 26.20 | 11.90 | 27.80 | 29.80 | 26.20 |
| | Forest R-CNN [36] | | 26.90 | 20.10 | 27.90 | 28.30 | 27.50 |
| | DropLoss [11] | | 26.90 | 14.80 | **29.80** | 28.30 | 26.80 |
| | RIO [1] | | 27.70 | 20.10 | 28.30 | 30.00 | 27.30 |
| | EQL v2 [29] | | 28.10 | **20.70** | 28.30 | **30.90** | **28.10** |
| | DisAlign [43] | | 25.80 | 10.30 | 27.60 | 29.60 | 25.60 |
| | RFS [8]⋆ | | 25.75 | 15.46 | 25.96 | 29.60 | 25.44 |
| | | ✓ | (+2.38) **28.13** | (+4.90) 20.36 | (+3.24) 29.20 | (+0.30) 29.90 | (+2.55) 28.00 |
| X-101 | Forest R-CNN [36] | | 28.50 | **21.60** | 29.70 | 29.70 | **28.80** |
| | RIO [1] | | 28.90 | 19.50 | 29.70 | 31.60 | 28.60 |
| | DisAlign [43] | | 27.40 | 11.00 | 29.30 | 31.60 | 26.80 |
| | RFS [8]⋆ | | 27.05 | 15.38 | 27.34 | 31.35 | 26.66 |
| | | ✓ | (+1.93) **28.98** | (+3.94) 19.32 | (+2.60) **29.94** | (+0.27) **31.62** | (+1.94) 28.60 |

a simple baseline such as RFS [8] to match or outperform all methods but Seesaw [32], which is trained with a stronger 2× schedule and an improved mask head. When paired with MosaicOS [42], NORCAL can achieve state-of-the-art performance with all different backbone models, suggesting that improving the feature (especially on rare objects) and calibrating the classifier are key ingredients to the success of long-tailed object detection and instance segmentation.

### C.2 Results on LVIS v0.5 Instance Segmentation

Many existing works focus on LVIS v0.5. In this subsection, we thus report the results of instance segmentation on LVIS v0.5 in Table B. Again, we observe similar trends that NORCAL can significantly improve the baseline models with all different backbone architectures. Particularly, we can also see improvements on the sigmoid-based object detector, *i.e.*, BALMS [24].

### C.3 Results on LVIS v0.5 Object Detection

In Table C, we compare with existing methods that reported results on LVIS v0.5 object detection — only the bounding box annotations are used for model training. Concretely, we include EQL [28], LST [12], BaGS [16], TFA [35], and MosaicOS [42], as the compared methods. In addition, we study a popular sigmoid-based detector, *i.e.*, RetinaNet with focal loss [19]. We train the RetinaNet using the default hyper-parameters [8] and apply NORCAL on top of it. We see that NORCAL can consistently improve the baseline models.

### C.4 Results on Objects365 dataset

We further validate NORCAL on Objects365 [26], a dataset designed to spur object detection research with a focus on diverse objects in the wild. Objects365 contains 2 million images, 30 million

Table C: **Object detection results on the validation set of LVIS v0.5.** NORCAL significantly boosts baseline methods. All models use Faster R-CNN [25] with ResNet-50 and FPN [18]. †: slight drop on frequent class. ♣: pre-trained with MSCOCO [17]. §: models trained by ourselves. ⋆: models from [42]. ‡: models from [16].

| Method | NORCAL | $AP^b$ | $AP^b_r$ | $AP^b_c$ | $AP^b_f$ |
|---|---|---|---|---|---|
| EQL [28] | | 23.30 | – | – | – |
| LST [12] | | 22.60 | – | – | – |
| BaGS [16]♣ | | 25.96 | 17.66 | 25.75 | 29.55 |
| RetinaNet [19]§ | | 16.34 | 9.47 | 14.07 | 21.93 |
| | ✓ | (+0.98) 17.32 | (+2.24) 11.71 | (+1.62) 15.69 | †(-0.32) 21.61 |
| Faster R-CNN [25]♣, ‡ | | 20.98 | 4.13 | 19.70 | 29.30 |
| | ✓ | (+2.89) 23.87 | (+2.85) 6.98 | (+4.47) 24.17 | (+0.94) 30.24 |
| RFS [8]⋆ | | 23.35 | 12.98 | 22.60 | 28.42 |
| | ✓ | (+2.27) 25.62 | (+4.57) 17.55 | (+2.93) 25.53 | (+0.53) 28.95 |
| TFA [35] | | 24.07 | 14.90 | 23.89 | 27.94 |
| | ✓ | (+0.56) 24.63 | (+1.72) 16.62 | (+0.84) 24.73 | †(-0.25) 27.70 |
| MosaicOS [42] | | 25.01 | 20.19 | 23.89 | 28.33 |
| | ✓ | (+2.53) 27.54 | (+4.88) **25.07** | (+3.32) 27.21 | (+0.60) 28.93 |
| MosaicOS [42]♣ | | 26.30 | 17.32 | 26.20 | 30.00 |
| | ✓ | (+2.05) **28.35** | (+5.82) 23.14 | (+2.19) **28.39** | (+0.37) **30.37** |

Table D: Results of object detection AP within each group of categories (according to the training image numbers) on Objects365 [26] validation set. The baseline model is Faster R-CNN with ResNet-50 and FPN.

| | AP | $AP_{(0,100)}$ | $AP_{[100,1000)}$ | $AP_{[1000,10000)}$ | $AP_{[10000,+\infty)}$ |
|---|---|---|---|---|---|
| # Category | 365 | 33 | 115 | 141 | 76 |
| Baseline | 16.29 | 2.43 | 6.95 | 20.88 | 27.93 |
| w/ NORCAL | (+0.48) 16.77 | (+0.23) 2.67 | (+0.54) 7.50 | (+0.48) 21.36 | (+0.50) 28.43 |

bounding boxes, and 365 categories with a long-tailed distribution. We train a Faster R-CNN [25] as the baseline on the training set, with FPN and ResNet-50 as the backbone. We report results in Table D. We not only show the overall mean AP, but also the mean APs for different groups of categories based on the training image number per category. NORCAL outperforms the baseline detector on all groups of categories, justifying its effectiveness and generalizability.

## C.5 Results on MSCOCO Dataset

We also experiment our method NORCAL on the generic object detection benchmark, *i.e.*, MSCOCO [17]. MSCOCO is the most popular benchmark for object detection and instance segmentation, which contains 80 categories with a relative balanced class distribution (See Figure A). More importantly, the least frequent class, "hair driver", still has 189 training images. In other words, all the classes in MSCOCO are considered as frequent classes using the definition of LVIS. We report results in Table E. We see that the performance gains brought by NORCAL is marginal. Our hypothesis is that the detectors trained with MSCOCO already see sufficient examples for all categories (even for tail classes) and the trained classifier is less biased.

Table E: **Results of object detection on MSCOCO [17].** The baseline model is from Faster R-CNN with FPN and ResNet-50 as the backbone.

| Method | AP | $AP_{50}$ | $AP_{75}$ | $AP_s$ | $AP_m$ | $AP_l$ |
|---|---|---|---|---|---|---|
| Baseline | 37.93 | 58.84 | 41.05 | 22.44 | 41.14 | 49.10 |
| w/ NORCAL | 37.96 | 58.40 | 41.22 | 22.22 | 41.18 | 49.48 |

## C.6 Results on Image Classification Datasets

Besides object detection and instance segmentation, we further evaluate NORCAL on three imbalanced classification benchmarks: ImageNet-LT [20], iNaturalist (2018 version) [31], and CIFAR-10-LT (with an imbalance factor 100) [3]. ImageNet-LT has 1,000 classes while iNaturalist has 8,142 classes. All three datasets have long-tailed distributions on the number of training images per class but have

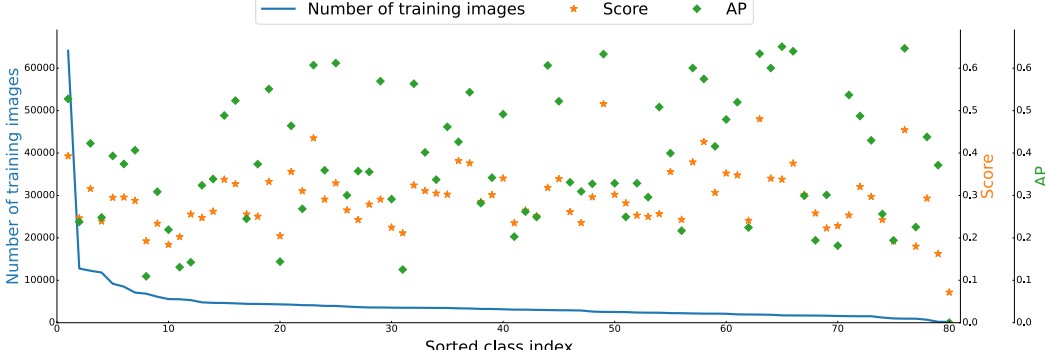

Figure A: **Per-class AP of Faster R-CNN and the category distribution on MSCOCO (2017).** The categories are sorted in descending numbers of training images. Orange stars indicate the average of predicted confidence scores for each class. Green diamonds are per-class APs. The least frequent class, "hair driver", still has 189 training images, indicating that all the classes in MSCOCO are considered as frequent classes using the definition of LVIS.

Table F: **Classification accuracy** on ImageNet-LT [20], iNaturalist [31], and CIFAR-10-LT [3].

| (a) ImageNet-LT | | | (b) iNaturalist | | | (c) CIFAR-10-LT | |
|---|---|---|---|---|---|---|---|
| Method | Top-1 | Top-5 | Method | Top-1 | Top-5 | Method | Top-1 |
| Baseline | 45.11 | 71.18 | Baseline | 61.54 | 82.94 | Baseline | 70.36 |
| w/ NORCAL | 49.71 | 74.53 | w/ NORCAL | 65.15 | 84.83 | w/ NORCAL | 77.78 |

a balanced evaluation set. We follow the literature to train a ResNet-50 classifier for the first two datasets, and a ResNet-32 classifier for CIFAR. Since there is no background class in these datasets, we simply drop the background class in Eq.4 in the main text. Results are shown in Table F. As expected, NORCAL consistently outperforms the baseline classifiers, demonstrating its effectiveness on long-tailed classification problems as well.

As mentioned in the Section 1 in the main paper, post-processing calibration for imbalanced or long-tailed classification has been studied in several prior works. Our approach is indeed inspired by their efficiency and effectiveness in classification problems and we extend them to the detection and instance segmentation problems.

## C.7 Ablation Studies on Sigmoid-Based Detectors (*i.e.*, with Multiple Binary Classifiers)

As shown in the main paper (cf. Table 3), we conduct ablation studies of NORCAL with a standard softmax-based object detection. Here, we further examine a sigmoid-based object detector, *i.e.*, BALMS [24], and report the results in Table G. Beyond Eq. 7 of the main paper, we ablate NORCAL with different calibration mechanisms, factors, and with and without score normalization. We note that, in this kind of models, $C$ binary classifiers are learned, each corresponds to one foreground class. In other words, no background class is specifically learned. Thus, the score normalization is usually not necessary or harmful — the background patches with low scores by all the classifiers will now gets their scores boosted due to calibration.

## C.8 Empirical Class Frequency is Better than Classifier Norms for NORCAL

As mentioned in the main paper (cf. Section 3.3 and Table 2 (bottom)), class-dependent temperature ($N_c^\gamma$) [38] provides a better signal for calibration than the classifier norms ($\|\boldsymbol{w}_c\|_2^\gamma$) of the classifier. Table H shows a comparison between those two factors for our proposed calibration mechanism. With NORCAL, we see that $N_c$ outperforms $\|\boldsymbol{w}_c\|_2$ on all object categories. Moreover, we notice that leaving the background intact shows a better performance, justifying our analysis and experimental results on how to handle the background class (cf. Section 3.2 and Figure 4 of the main paper).

Table G: **Ablation studies of NORCAL with the sigmoid-based baseline model (BALMS [24]).** We follow Ren et al. [24] to report the results on LVIS v0.5 instance segmentation. CAL: calibration mechanism. NOR: class score normalization. The best ones are in bold. As discussed in Section C.7, normalization is not suitable for this kind of models.

| $a_c$ | CAL | NOR | AP | $AP_r$ | $AP_c$ | $AP_f$ |
|---|---|---|---|---|---|---|
| Baseline | $\exp(-\phi_c(\boldsymbol{x}))$ | ✓ | 26.97 | 17.31 | 28.07 | **29.47** |
| $\dfrac{1-\gamma^{N_c}}{1-\gamma}$ (ENS [3]) | $\exp(-\phi_c(\boldsymbol{x}) \times a_c)$ | ✗ | 26.99 | 17.40 | 28.06 | 29.46 |
| | | ✓ | 15.56 | 7.73 | 14.91 | 19.51 |
| | $s_c \times a_c$ | ✗ | 27.12 | **19.89** | 28.25 | 28.59 |
| | | ✓ | 15.29 | 12.05 | 16.98 | 14.47 |
| | $\exp(-\phi_c(\boldsymbol{x})) \times a_c$ | ✗ | 27.17 | 19.88 | 28.26 | 28.71 |
| | | ✓ | 18.62 | 12.34 | 18.29 | 21.55 |
| $N_c^{\gamma}$ (CDT [38]) | $\exp(-\phi_c(\boldsymbol{x}) \times a_c)$ | ✗ | 27.37 | 18.64 | 28.69 | 29.22 |
| | | ✓ | 16.82 | 9.50 | 17.24 | 19.22 |
| | $s_c \times a_c$ | ✗ | 27.52 | 19.33 | **28.82** | 29.17 |
| | | ✓ | 15.62 | 11.58 | 17.32 | 15.10 |
| | $\exp(-\phi_c(\boldsymbol{x})) \times a_c$ | ✗ | **27.52** | 19.34 | 28.80 | 29.19 |
| | | ✓ | 18.60 | 12.64 | 18.36 | 21.28 |

Table H: **Empirical class frequency ($N_c$) is better than classifier norms ($\|\boldsymbol{w}_c\|_2$) for NORCAL.** Results are reported on LVIS v1 instance segmentation. Background: whether calibrating the background class or not.

| Method | $a_c$ | Background | AP | $AP_r$ | $AP_c$ | $AP_f$ | $AP^b$ |
|---|---|---|---|---|---|---|---|
| RFS [8] | – | – | 22.58 | 12.30 | 21.28 | 28.55 | 23.25 |
| w/ NORCAL | $\|\boldsymbol{w}_c\|_2^{\gamma}$ | ✗ | 22.86 | 13.21 | 21.67 | 28.43 | 23.41 |
| | | ✓ | 22.56 | 12.47 | 21.34 | 28.37 | 23.17 |
| | $N_c^{\gamma}$ | ✗ | 25.22 | 19.33 | 24.16 | 28.98 | 26.08 |

## C.9 Further Analysis on Existing Post-Processing Calibration Methods

We compare NORCAL to the existing post-calibration methods in the main paper (cf. Table 2 (upper)). In the main paper, we follow the implementations in [4] to perform the calibration after the top 300 predicted boxes are selected. Here we study an alternative of directly applying the calibration before selecting the 300 predictions. We show the results in Table I. NORCAL still outperforms all existing calibration methods.

Table I: **Further analysis and comparison on existing post-processing calibration methods.** Results are reported on LVIS v1 instance segmentation. When to calibrate: before or after the top 300 predicted boxes are selected per image.

| Method | When to calibrate? | AP | APr | APc | APf |
|---|---|---|---|---|---|
| RFS [8] | – | 22.58 | 12.30 | 21.28 | 28.55 |
| w/ HistBin [39] | before | 18.91 | 5.65 | 17.49 | 26.33 |
| | after | 21.82 | 11.28 | 20.31 | 28.13 |
| w/ BBQ (AIC) [21] | before | 16.56 | 3.07 | 14.76 | 24.51 |
| | after | 22.05 | 11.41 | 20.72 | 28.21 |
| w/ Beta calibration [14] | before | 22.11 | 11.54 | 21.77 | 27.15 |
| | after | 22.55 | 12.29 | 21.27 | 28.49 |
| w/ Isotonic seg. [40] | before | 20.58 | 10.46 | 20.36 | 25.27 |
| | after | 22.43 | 12.19 | 21.12 | 28.41 |
| w/ Platt. scaling [22] | before | 22.09 | 12.07 | 21.40 | 27.26 |
| | after | 22.55 | 12.29 | 21.27 | 28.49 |
| w/ NORCAL | before | 25.22 | 19.33 | 24.16 | 28.98 |

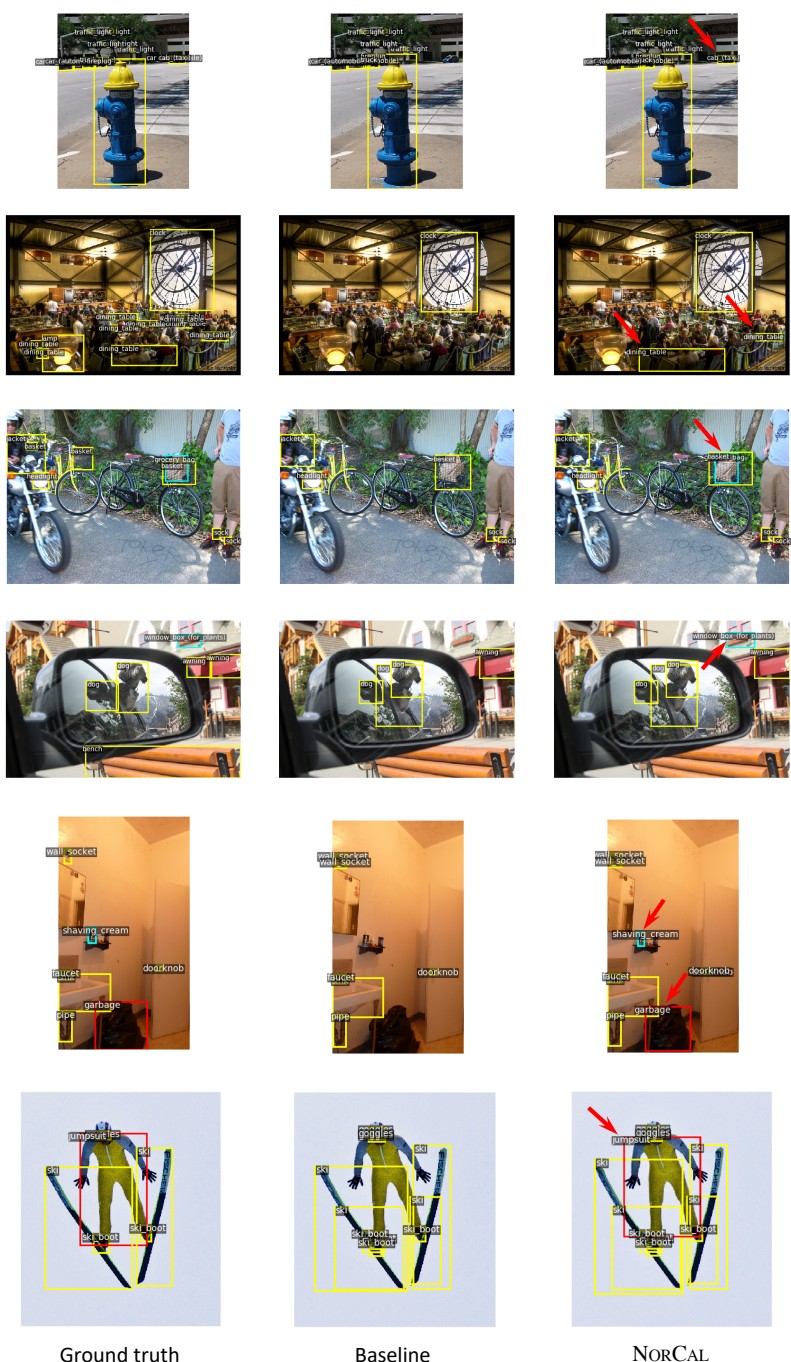

Ground truth          Baseline          NORCAL

Figure B: **Additional qualitative results.** We superimpose red arrows to show the improvement. Yellow, cyan and red bounding boxes indicate frequent, common and rare class labels.

## C.10    Additional Qualitative Results

We provide additional qualitative results on LVIS v1 in Figure B. We show the (predicted) bounding boxes from the ground truth annotations, the baseline Mask R-CNN [10] with RFS [8], and NORCAL.