# OpenReview forum: "On Model Calibration for Long-Tailed Object Detection and Instance Segmentation"
_NeurIPS.cc/2021/Conference — NeurIPS 2021 Poster_

### Official Review · Reviewer_PQBv · 2021-07-13

**Rating:** 7
**Confidence:** 4

**Summary:**

In this paper, the authors study the problem of "calibration" of long-tail object detection models. To be specific, they take an existing pre-trained object detector and modify its logits according to the number of samples in each class to obtain better logits/scores for each class. They demonstrate on the long-tailed LVIS dataset with different models that this significantly and consistently improves the performance of methods (in terms of both precision and recall).

**Limitations And Societal Impact:**

Yes, they have.

**Main Review:**

Strengths:
- A simple yet robust and generic method.
- Extensive ablation analysis.
- Strong results.
- Clear and easy to follow text.

Weaknesses:
- The paper provides a very simple application of the normalization methods in other studies for object detection. In this sense, technical/methodological contribution is limited. Though, I favor simple solutions and I am not going to reflect this in my recommendation.

- This is a paper for addressing an imbalance problem for object detection and therefore, the authors should cite the following review paper on imbalance problems in object detection:

Oksuz et al., "Imbalance Problems in Object Detection: A Review", IEEE Transactions on Pattern Analysis and Machine Intelligence (PAMI), 2021.

- Figure 2: It is not clear how these scores are obtained. Which split of the dataset is used? Which detector is used? How did you perform normalization for the plot on the left? Are the scores on the right not normalized? Why not? What do you mean by "another rare" classes? ..

Minor comments:
- Figure 2: It would be nicer if the font sizes & styles of the text in the figure match the font size and the style of the main text. Moreover, it is very clear that the spacing between the figure and the text has been played with. This is against the formatting instructions.
- Footnote 2 on page 4: The period (.) is missing.
- Line 133: N_c has not been introduced yet.
- Line 136: "confidential scores" => "confidence scores".
- Line 148: "posts" => "posits".
- Line 149: "background patches" => "background regions".
- Line 231: "methods that trained" => "methods that are trained".

**AFTER REBUTTAL**

I've looked at the comments of the other reviewers and noticed that we are generally very positive about the paper. Some of us requested more experiments and the authors provided the results for them in their rebuttal, which supported the claims of the authors. Therefore, I will keep my original score of acceptance for the paper.

**Time Spent Reviewing:**

2.5

---

> ### Author Response · Authors · 2021-08-10
> **To Reviewer PQBv**
>
> Thanks for the encouraging review and the detailed feedback.
>
> **Missing reference:** Thanks and we will include it in the final version.
>
> **Figure 2:** Sorry for not making it clear. Figure 2 is based on the baseline Faster R-CNN model on the LVIS v0.5 validation set. For the left figure, we studied the confidence scores of the top 300 tuples (Line 123-129) of each image by Fast R-CNN w/o or w/ NorCal. We average them for rare, common, and frequent classes and then linearly scale these averaged scores such that the frequent class has a score of 1. The normalization here is just to make the score comparison among classes clearer; it is not the normalization applied in NorCal. We will clarify this in the final version.
>
> For the right figure, we only studied the baseline Faster R-CNN; thus, we do not apply the above-mentioned normalization. As mentioned in Line 123-126, each detected tuple is formed by an object proposal, a predicted class, and the confidence score of that class. In Faster R-CNN and many other detectors, each proposal can be paired with multiple predicted classes and appears in multiple tuples if the corresponding scores are high enough, as noted in footnote 1. Hence, for each tuple predicted as a rare class, we look at its proposal and further show the highest score of *another* rare, common, and frequent class from the *same* proposal. The *another* here means beyond the predicted rare class. The right figure indicates that for each object proposal, the learned baseline detector will tend to first classify it as a frequent class. That is, rare objects are usually misclassified into a frequent class.
>
>
> **Other comments:** Thank you and we will include/correct them in the final version.

---

### Official Review · Reviewer_aBnL · 2021-07-16

**Rating:** 6
**Confidence:** 4

**Summary:**

This paper presents a simple yet effective approach “Normalized Calibration” to post-process the class-specific logits from modern object detector using soft-max prediction for all object and one background classes. Experiments on LVIS has been conducted, and comparison with quite a few baseline methods are presented to show its superior performance for frequent, common and rare classes.

**Ethical Concerns:**

I do not have Ethical Concerns.

**Limitations And Societal Impact:**

I do not have concerns on the negative societal impact of this work.

**Main Review:**

The idea seems to be original, which is to scale down the exponentialized logits by the class size in the training set and L1-normalize scores. Without L1-normalization, the calibration has a similar form to tar-normalization in [20] (e.g, scale down the logits before taking exponentiation) (also see table 3). It seems L1-normalization plays a more important role than logit calibration.

The paper writing is clean and easy to follow. External comparisons and internal ablative studies are convincing. There are enough clarity in the technical exposition.

One limitation is NorCal seems to work less well for detectors predicting sigmoid scores that are not necessarily L1-normalized (e.g. RetinaNet with focal loss). I would like to see more results using detectors using sigmoid rather than softmax to compute class scores.

Currently, only one task and one dataset LVIS is used for evaluation. The proposed NorCal seems to be generic enough to be evaluated on long-tailed classification such as ImageNet-LT and iNaturalist benchmark. Results will be more convincing if results on a 2nd dataset or task are provided.

**Time Spent Reviewing:**

4 hrs

---

> ### Author Response · Authors · 2021-08-10
> **To Reviewer aBnL**
>
> We thank the reviewer for the valuable comments.
>
> **Calibration vs. normalization:** We would like to emphasize that both score calibration and normalization are essential to the success of NorCal. The former re-ranks the confidence scores *across classes* to overcome the bias that rare classes usually have lower confidence scores; the latter helps re-order the scores of detected tuples *within each class* to further improve the performance. In addition to normalization, our approach scales the logits differently from tau-normalization [20]. We scale the logits based on class sizes rather than the classifier norms (section 3.3). We conducted a detailed ablation study on these two methods in section C.6 and Table F in the supplementary material. Even with normalization, the results by tau-normalization are still much lower than our approach, demonstrating that how to scale the logits is important.
>
>
> **Sigmoid-based detectors:** Thanks for the comment. We have studied two sigmoid-based detectors in the paper, i.e., EQL (Line 231-232, 240-241, and Table 1) and BALMS (Table B in the supplementary material). NorCal can notably improve the performance of rare and common classes. As suggested, we further train a RetinaNet with the focal loss on LVIS v0.5 for long-tailed object detection, using the default hyperparameters. As shown below, NorCal can again improve upon it, especially for the rare and common classes. Besides these results, we have also conducted a detailed ablation study on NorCal for sigmoid-based detectors in section C.5 of the supplementary material.
>
> **Table.** AP on LVIS v0.5 validation set with RetinaNet.
>
> |             |    AP |  APr |  APc |   APf |
> |------------:|------:|-----:|-----:|------:|
> |    Baseline |  16.34 | 9.47 | 14.07 | 21.93 |
> |   w/ NorCal | 17.32 | 11.71 | 15.69 | 21.61 |
> | Improvement |  +0.98 | +2.24 | +1.62 |  -0.32 |
>
>
> **Results on other tasks and datasets:** Thanks for the suggestion. Please refer to the response ***Other datasets*** to *Reviewer bKAV* for an additional dataset Objects365 on long-tailed detection. Please also refer to the response  ***Classification results*** to *Reviewer 3gEu* for the classification results on ImageNet-LT and iNaturalist. NorCal can consistently improve the baseline detections and classifiers in these experiments, demonstrating its effectiveness and generalizability.

---

### Official Review · Reviewer_3gEu · 2021-07-16

**Rating:** 5
**Confidence:** 4

**Summary:**

A simple and effective post-processing calibration method is proposed for long-tailed object detectors.

**Limitations And Societal Impact:**

Yes

**Main Review:**

Strength
- The paper presents an interesting idea. It is quite effective and requires no re-training of the object detectors, making it a useful and low-effort tool to use. Experiments also show superior performance than other methods which requires a change of training.
- The experiments revealed that the key to performance improvement is normalization especially for the frequent classes, which is an important result and helps us better understand how the method works.
- The paper is well written and clear. The illustrations to explain why the proposed method works are very useful.

Weakness
- The proposed method is purely about post-processing of classification scores in object detectors. Thus the localization part is not important. Therefore it makes sense to start by studying the long-tailed image classification problem and then apply it to object detection/instance segmentation problem.  It'll be better if the authors can show the efficacy of this method on classification problems as well.
- It makes sense that the proposed method works well with multi-class classifiers. However, for multiple binary classifiers, as described in Section 3.4 and Eq. 7, there's no normalization applied. In this case, the ranking of the boxes per class won't be changed because Eq. 7 is a monotonic function. So how is the performance improvement achieved?
- Although the proposed method works well in the experiments, the overall novelty is not high.

Minor comments
- In Eq. 7, should it be dividing by the factor instead of multiplying it?

**Time Spent Reviewing:**

3

---

> ### Author Response · Authors · 2021-08-10
> **To Reviewer 3gEu**
>
> We thank the reviewer for the valuable comments.
>
> **Classification results:** Thanks for the suggestion. We evaluate the post-processing calibration technique of NorCal on three benchmark imbalanced classification datasets: ImageNet-LT [A], iNaturalist (2018 version) [B], and CIFAR-10-LT (with an imbalance factor 100) [C]. ImageNet-LT has 1,000 classes while iNaturalist has 8,142 classes. All three datasets have long-tailed distributions on the number of training images per class but have a balanced evaluation set. We follow the literature to train a ResNet-50 classifier for the first two datasets, and a ResNet-32 classifier for CIFAR. Since there is no background class in these datasets, we simply drop the background class in Equation 4. Results are shown below. NorCal consistently outperforms the baseline classifiers, demonstrating its effectiveness on long-tailed classification problems as well.
>
> We note that post-processing calibration for imbalanced or long-tailed classification has been studied in several prior works, as mentioned in Line 30-35, 78-82, and 159. Our approach is indeed inspired by their efficiency and effectiveness in classification problems and we extend them to the detection and instance segmentation problems (Line 25-29, 36-38).
>
> **Table.** The classification accuracy on ImageNet-LT [A],  iNaturalist [B], and CIFAR-10-LT [C].
>
> |             |    ImageNet-LT |                |    iNaturalist |                |    CIFAR-10-LT |
> |------------:|---------------:|---------------:|---------------:|---------------:|---------------:|
> |             | Top-1 accuracy | Top-5 accuracy | Top-1 accuracy | Top-5 accuracy | Top-1 accuracy |
> |    Baseline |          45.11 |          71.18 |          61.54 |          82.94 |          70.36 |
> |   w/ NorCal |          49.71 |          74.53 |          65.15 |          84.83 |          77.78 |
> | Improvement |           +4.60 |           +3.35 |           +3.61 |           +1.89 |           +7.42 |
>
> [A] Liu, Ziwei, Zhongqi Miao, Xiaohang Zhan, Jiayun Wang, Boqing Gong, and Stella X. Yu. "Large-scale long-tailed recognition in an open world." In CVPR, 2019.
>
> [B] Van Horn, Grant, Oisin Mac Aodha, Yang Song, Yin Cui, Chen Sun, Alex Shepard, Hartwig Adam, Pietro Perona, and Serge Belongie. "The inaturalist species classification and detection dataset." In CVPR, 2018.
>
> [C] Cui, Yin, Menglin Jia, Tsung-Yi Lin, Yang Song, and Serge Belongie. "Class-balanced loss based on effective number of samples." In CVPR, 2019.
>
> **Improvement on multi binary classifiers:** We note that the AP is affected by two factors. First, for an object to be detected *within an image* (i.e., true positive), it must have a confidence score large enough to be included in the cap (Line 127-129). Such a cap makes sense in practice since a scene seldom contains a large number of objects. Creating too many, likely noisy detection tuples can be annoying to users as well (e.g., for a camera equipped with an object detector). Second, given that an object is correctly detected, re-ordering it *within each class* to have a higher rank than other false positives would further improve the AP. For multiple binary classifiers, score calibration alone can re-rank the detected tuples across classes *within each image* such that rare and common objects, which initially have lower scores (Line 44-45, 136-138), could be included in the cap to largely increase the recall. As a result, the improvement for multiple binary classifiers mainly comes from the rare and common objects, with a light degradation on the frequent classes.
>
>
> **Novelty and contributions:** We want to reiterate that our main contribution and novelty is in demonstrating that with careful design, the simple post-processing calibration can efficiently and effectively improve the long-tailed object detection results. We conducted extensive and rigorous experiments to show that such a technique is compatible with many existing solutions so that we can further improve the state-of-the-art performance. We also provided illustrations and analysis to explain the improvement made by our approach, as mentioned by the reviewer as the paper’s strength. We believe that the simplicity and wide applicability of our approach make it a valuable contribution.

---

### Official Review · Reviewer_bKAV · 2021-07-16

**Rating:** 6
**Confidence:** 4

**Summary:**

This paper presents a normalized calibration method for long-tailed object detection and instance segmentation task. Experimental resutls show that separately treating the background class and normalizing the scores over classes for each proposal are keys to achieving superior performance. Experiments are conducted on the LVIS dataset with diverse evaluation sets. Results shows that consistent improvement can be achieved by the proposed scheme, and  extensive analysis and ablation studies are provided.


**Limitations And Societal Impact:**

Authors doesn't address the limitation and potential negative societal impact.

**Main Review:**

The paper apply simple post-calibration techniques from standard multi-way to object detection and instance segmentation. The main idea is to scale down the logit of each class with respect to its train size.
Overall, the paper is well organized, and the proposed scheme is clearly presented. The idea is straightforward and easy to implement, as only Hyper-parameter is introduced for tune in the scheme. The extensive experiments offer insights into various modeling choices and mechanisms of the scheme.
It would be better to conduct experiments on a more general dataset, (e.g. Open Images) to demonstrate its effectiveness.


**Time Spent Reviewing:**

3 hours

---

> ### Author Response · Authors · 2021-08-10
> **To Reviewer bKAV**
>
> We thank the reviewer for the positive feedback.
>
> **Other datasets:** Thanks for the suggestions. Given limited time, we experiment on Objects365 [A], a dataset designed to spur object detection research with a focus on diverse objects in the wild. Objects365 contains 2 million images, 30 million bounding boxes, and 365 categories with a *long-tailed* distribution. We train a Faster R-CNN as the baseline on the training set, with FPN and ResNet-50 as the backbone. We not only show the overall mean AP, but also the mean APs for different groups of categories based on the training image number per category. NorCal outperforms the baseline detector on all groups of categories, justifying its effectiveness and generalizability. We will further experiment on Open Images in the final version. We also note that, in Section C.4 of the supplementary material, we have also applied NorCal to MSCOCO. Since MSCOCO is a relatively balanced dataset, as expected we did not observe a notable improvement.
>
> **Table.** AP within each group of categories (according to the training image numbers) on Objects365 [A] validation set.
>
> |    | All | (0, 100) | [100, 1000) | [1000, 10000) | [10000, +$\infty$) |
> |---:|---:|---:|---:|---:|---:|
> | #Classes | 365 | 33 | 115 | 141 | 76 |
> | Baseline | 16.29 | 2.43 | 6.95 | 20.88 | 27.93 |
> | w/ NorCal | 16.77 | 2.67 | 7.50 | 21.36 | 28.43 |
> | Improvement | +0.48 | +0.23 | +0.54 | +0.48 | +0.50 |
>
>
> [A] Shao, Shuai, Zeming Li, Tianyuan Zhang, Chao Peng, Gang Yu, Xiangyu Zhang, Jing Li, and Jian Sun. "Objects365: A large-scale, high-quality dataset for object detection." In CVPR, 2019.
>
>
> **Limitations & societal impact:** We provided some discussions in Sections E and D of the supplementary material, and we will clearly indicate them or move them into the main text in the final version.

---

### Author Response · Authors · 2021-08-10
**General response**

We thank reviewers for their valuable comments. We are glad that they find our idea "straightforward and easy to implement" (R1), "interesting" (R2), "original" (R3); our approach "simple" (R1, R4), "effective" (R2, R3), "useful" (R2), "robust and generic" (R4); our results "superior", "strong" (R1, R2, R3, R4); our experiments "extensive" (R1, R4), "convincing" (R3), "important" (R2); our paper "well-written and clear" (R1, R2, R3, R4). We separately address each reviewer’s comments as follows. We will incorporate all the feedback into the final version.

R1: Reviewer bKAV; R2: Reviewer 3gEu; R3: Reviewer aBnL; R4: Reviewer PQBv.

---

> ### Author Response · Authors · 2021-08-22
> **To all reviewers: please let us know if you have further questions**
>
> Dear reviewers and AC,
>
> Thank you for reading our rebuttal.
>
> We have tried to address all concerns raised by the reviewers. Please let us know if you have any further questions about our paper or rebuttal. Thank you.

---

> > ### Author Response · Authors · 2021-08-26
> > **To all reviewers: please let us know if you have further questions**
> >
> > Dear reviewers and AC,
> >
> > Thank you for reading our rebuttal.
> >
> > We have tried to address all concerns raised by the reviewers. Please let us know if you have any further questions about our paper or rebuttal. Thank you.

---

### Author Response · Authors · 2021-08-31
**To all reviewers: please let us know if you have further questions**

Dear reviewers and AC,

Thank you for reading our paper and rebuttal. We have tried to address all concerns raised by the reviewers. We appreciate that Reviewer PQBv has provided positive feedback to our rebuttal. Reviewer PQBv also said, "I've looked at the comments of the other reviewers and noticed that we are generally very positive about the paper."

Please let us know if you have any further questions that we could clarify for you to revise your score. Thank you.

---

### Decision · Program_Chairs · 2021-09-27

**Decision:**

Accept (Poster)

**Comment:**

The reviewers were generally positive about this paper, and most of the raised concerns were cleared after the responce. Thus I recommend accepting the paper.